# EFFICIENT ITERATIVE POLICY OPTIMIZATION

**Nicolas Le Roux**
Criteo Research
nicolas@le-roux.name

## ABSTRACT

We tackle the issue of finding a good policy when the number of policy updates is limited. This is done by approximating the expected policy reward as a sequence of concave lower bounds which can be efficiently maximized, drastically reducing the number of policy updates required to achieve good performance. We also extend existing methods to negative rewards, enabling the use of control variates.

## 1 INTRODUCTION

Recently, reinforcement learning has seen a surge in popularity, in part because of its successes in playing Atari games (Mnih et al., 2013) and Go (Silver et al., 2016). Due to its ability to act in settings where the actions taken influence the environment and, more generally, the input distribution of examples, reinforcement learning is now used in other domains, such as online advertising.

The goal is to learn a good policy, i.e. a good mapping from states to actions, which will maximize the final score, in the case of Atari games, the probability of winning, in Go, or the number of sales, in online advertising. We have at our disposal logs of past events, where we know the states we were in, the actions we took and the resulting rewards we obtained. In this paper, we shall focus on how to efficiently use these logs to obtain a good policy. In some cases, in addition to (state, action, reward) triplets, we have access to a teacher which provides the optimal, or at least a good, action for a given state. The use of such a teacher to find a good initial policy is outside the scope of this paper.

There are many ways to learn good policies using past data. The two most popular are Q-learning and direct policy optimization. In Q-learning (Sutton & Barto, 1998), we are trying to learn a mapping from a (state, action) pair to the reward. Given this mapping and a state, we can then find the action which leads to the maximal predicted reward. This method has been very successful, especially when the action space is small, since we need to test all the actions, and the reward somewhat predictable, since taking the maximum is unstable and a small error can lead to suboptimal actions.

Direct policy optimization, rather than trying to estimate the value of a (state, action) pair, directly parameterizes a policy, i.e. a conditional distribution over actions given the current state. More precisely, and using the notation from Kober (2014), we wish to maximize the expected return of a policy $p(\cdot|\theta)$ with parameters $\theta$, i.e.

$$J(\theta) = \int_{\mathcal{T}} p(\tau|\theta)R(\tau)\,d\tau\,, \qquad (1.1)$$

where $\mathcal{T}$ is the set of all paths $\tau$ and $R(\tau)$ is the reward associated with path $\tau$.

A rollout $\tau = [s_{1:T+1}, a_{1:T}]$ is a series of states $s_{1:T+1} = [s_1, \ldots, s_{T+1}]$ and actions $a_{1:T} = [a_1, \ldots, a_T]$. $p(\tau|\theta)$ is the probability of rollout $\tau$ when using a policy with parameters $\theta$ and $R(\tau)$ is the aggregated return of $\tau$. If we make the Markov assumption that a state only depends on the previous state and the action chosen, we have $p(\tau|\theta) = p(s_1) \prod_{t=1}^{T} p(s_{t+1}|s_t, a_t)\pi(a_t|s_t, t, \theta)$ where $p(s_{t+1}|s_t, a_t)$ is the next state distribution and is independent of our policy. The action space can be discrete or continuous.

Without the ability to exactly compute $J$[a], we must resort to sampling to get an estimate of both $J$ and its gradient. These samples can come from $p(\cdot|\theta)$ or from another distribution. In the latter case, we need to use importance sampling to keep our estimator unbiased[b]. Whether they use importance sampling or not, all methods which directly optimize the policy rely on iterative procedures. First, rollouts are performed under a policy to gather samples. Then, these samples are used to update the policy, which is in turn used to gather new samples, until convergence of the policy. These methods continuously gather new samples, mostly because the updates to the policy are more reliable when they are based on fresh samples. However, in a production environment, as we will see in Sec. 2, we will typically release a new policy to many users at once, gathering millions or billions of samples, but the delay between two updates of a policy is of the order of hours or even days. Thus, there is a strong need for policy learning techniques which can achieve a good performance with a limited number of policy updates. There is an analogous issue in robotics where each new rollout induces wear and tear on the robot, making such a method which limits the number of rollouts desirable.

In this paper, we shall thus present a method dedicated to achieving high performance while limiting the number of different policies under which samples are gathered. Sec. 3 reviews the relevant literature. Sec. 4 presents a first version of our algorithm, proving its theoretical efficiency and providing a common framework to several existing techniques. Then, Sec. 5 shows how the positiveness assumption for the rewards can slow down learning and proposes an improvement which circumvents the issue. Sec. 6 shows results of the proposed algorithm on both a synthetic dataset and a real-life, large scale dataset in online advertising. Finally, Sec. 7.3 reflects on the current state and the future venues of research.

## 2 DISPLAY ADVERTISING

Retargeting is a type of advertising where ads are displayed to users who have already expressed interest in one or multiple products, generally by browsing on merchants' websites. Since the information used to know which ad to display are not related to a query, like in search advertising, these ads can be displayed on any website, for instance news sites or personal pages. More precisely, every time a user lands on such a website, the website contacts an ad-exchange platform which runs a real-time auction. The highest bidder gets the right to display an ad for this particular user and pays a price depending on many factors, including the opponents' bids. If the user then clicks on the ad, the retargeter is paid by the merchant whose ad was displayed. To maximize its revenue, the retargeter must thus only display ads leading to a click, and do so at the lowest possible price. Historically, these auctions were second-price, which means that the price paid by the highest bidder was the second highest bid. From a bidder perspective, the optimal strategy was straightforward as the optimal bid was the expected gain of displaying an ad. However, ad-exchanges have recently moved to other types of auctions, where the optimal strategy depends on the (unknown) bids of the other bidders. Worse, the exact type of auction is unknown to the bidders who only know the price they pay when they win the auction.

The bidding problem thus fits nicely in the reinforcement learning framework where the state is the set of information known about the user and the current website, the action is the bid and the reward is the payment (if there is a click) minus the cost of displaying the ad. As the reward depends mostly on whether there is a click or not, an event which is highly unpredictable, techniques such as doubly robust estimation (Dudík et al., 2011) or based on carefully crafted Q-functions (Lillicrap et al., 2015; Schulman et al., 2015b; Gu et al., 2016) are unlikely to yield significant improvements.

There is another major difference with other reinforcement learning works. For quality control, a new policy can only be put in production every few hours or even days. Since large advertising companies display several billion ads each day, each new policy is used to gather about a billion samples. We are thus in a very specific setting where the number of samples is very large but the number of policies with which samples are gathered is limited.

We will now review the existing literature and show how no existing work addresses the constraints we face. We will then present our solution which is both simple and leads to good policies. To

---

[a]Computing $J$ would require visiting every possible $\tau$ at least once, which is impossible, even for moderately long rollouts.

[b]Using a biased estimator can be useful but this is outside the scope of this paper.

show its efficiency, we report results on both the Cartpole problem, a synthetic problem common in reinforcement learning, and a real-world example.

## 3    RELATED WORK

We review here some of the most common techniques in reinforcement learning, limiting ourselves to those who try to directly optimize the policy.

The first such method is REINFORCE (Williams, 1992), which performs a single gradient step between two rollouts of the policy. This method has multiple issues. First, one has to do rollouts after each update, ultimately resulting in many rollouts before reaching a good solution. This is further emphasized with the potential poor quality of the gradient update which is not insensitive to a reparametrization of the parameter space. Finally, as with any stochastic method, the choice of the stepsize has a strong influence of the convergence speed. Each of these problems has been treated in separate works. The need to perform rollouts after each update was alleviated by using importance sampling policy gradient (Swaminathan & Joachims, 2015). The update direction can be improved by using natural gradient (Amari, 1998; Kakade, 2001) and doing a line search helps in choosing a correct stepsize (Jie & Abbeel, 2010). These improvements can be computationally expensive and the additional hyperparameters make them less suited to a production environment where simplicity and robustness are key.

Another line of research used concave lower bounds on $J(\theta)$, which could then be optimized using off-the-shelf classifiers such as L-BFGS (Liu & Nocedal, 1989). Examples of such methods are PoWER (Kober & Peters, 2009) and Natural actor-critic (Peters & Schaal, 2008). These bounds were obtained using an analogy with EM (Dempster et al., 1977; Minka) which required the rewards $R(\tau)$ to be positive. In settings where multiple policies can achieve high accuracy, Neumann (2011) proposed another EM-based method which focuses on one of these policies rather than trying to loosely cover all of them, at the expense of a larger computational cost.

We will show how these works can be extended to better optimize the policy between two updates. We will then show how the positiveness requirement hurts the optimization, then propose an extension which allows us to use any reward.

Since, in practice, we do not have access to the full distribution but rather to a set of $N$ samples, we shall optimize a Monte-Carlo estimate of $J$:

$$\hat{J}(\theta) = \frac{1}{N} \sum_i R(\tau_i) \frac{p(\tau_i|\theta)}{p(\tau_i|\theta_0)} \quad , \quad \tau_i \sim p(\tau|\theta_0) \; , \tag{3.1}$$

where $p(\tau|\theta_0)$ is the probability of rollout $\tau$ under the distribution used to generate samples. This is the standard importance sampling trick commonly used in policy gradient (Sutton et al., 1999).

## 4    CONCAVE APPROXIMATION OF THE EXPECTED LOSS

In this section, we assume that all returns are nonnegative[c]. Due to this nonnegativity, the non-concavity in $J$ stems from the nonconcavity of each $p(\tau|\theta)$. However, if $p(\tau|\theta)$ belongs to the exponential family (Wainwright & Jordan, 2008), then it is a log-concave function of its parameters and $\log p(\tau|\theta)$ is a concave function of $\theta$. This suggests the following lower bound:

**Lemma 1.** *Let*

$$p_q(\tau|\theta) = q(\tau) \left( 1 + \log \frac{p(\tau|\theta)}{q(\tau)} \right) \tag{4.1}$$

*with q such that* $q(\tau) \neq 0$ *when* $p(\tau|\theta) \neq 0$*. Then we have* $p_q(\tau|\theta) \leq p(\tau|\theta)$*.*

---

[c]Or at least bounded below, in which case they need to be adequately shifted.

*Proof.*

$$p(\tau|\theta) = q(\tau)\frac{p(\tau|\theta)}{q(\tau)}$$

$$\geq q(\tau)\left(1 + \log\frac{p(\tau|\theta)}{q(\tau)}\right)$$

$$= p_q(\tau|\theta) \ .$$

The second line stems from the inequality $x \geq 1 + \log x$. □

Lemma 1 shows that, regardless of the function $q$ chosen, $p_q(\tau|\theta)$ is a lower bound of $p(\tau|\theta)$ for any value of $\theta$. Thus, provided that $p(\tau|\theta)$ belongs to the exponential family, we have obtained a log-concave lower bound. Lemma 1, however, does not guarantee the quality of that lower bound. This is addressed by the following lemma:

**Lemma 2.** *If there is a $\nu$ such that $q(\tau) = p(\tau|\nu)$, we have*

$$p_q(\tau|\nu) = p(\tau|\nu) \quad , \quad \left.\frac{\partial p_q(\tau|\theta)}{\partial\theta}\right|_{\theta=\nu} = \left.\frac{\partial p(\tau|\theta)}{\partial\theta}\right|_{\theta=\nu} \ .$$

*Proof.* $p_q(\tau|\nu) = p(\tau|\nu)$ is immediate when setting $\theta = \nu$ in Eq. 4.1. Deriving $p_q(\tau|\theta)$ with respect to $\theta$ yields

$$\frac{\partial p_q(\tau|\theta)}{\partial\theta} = p(\tau|\nu)\frac{\partial\log p(\tau|\theta)}{\partial\theta}$$

$$= \frac{p(\tau|\nu)}{p(\tau|\theta)}\frac{\partial p(\tau|\theta)}{\partial\theta} \ .$$

Taking $\theta = \nu$ on both sides yields $\left.\frac{\partial p_q(\tau|\theta)}{\partial\theta}\right|_{\theta=\nu} = \left.\frac{\partial p(\tau|\theta)}{\partial\theta}\right|_{\theta=\nu}.$ □

To simplify further notations, we will write directly

$$p_\nu(\tau|\theta) = p(\tau|\nu)\left(1 + \log\frac{p(\tau|\theta)}{p(\tau|\nu)}\right) \tag{4.2}$$

to explicitly show the dependency of the bound on the parameter $\nu$.

The following result is a direct consequence of these two lemmas:

**Lemma 3.** *(lower bound of the expected reward estimator): Let*

$$\hat{J}_\nu(\theta) = \frac{1}{N}\sum_i R(\tau_i)\frac{p(\tau_i|\nu)}{p(\tau_i|\theta_0)}\left(1 + \log\frac{p(\tau_i|\theta)}{p(\tau_i|\nu)}\right) \ . \tag{4.3}$$

*Then we have $\hat{J}_\nu(\theta) \leq \hat{J}(\theta) \ \forall\theta \quad , \quad \hat{J}_\nu(\nu) = \hat{J}(\nu) \quad , \quad \left.\frac{\partial\hat{J}_\nu(\theta)}{\partial\theta}\right|_{\theta=\nu} = \left.\frac{\partial\hat{J}(\theta)}{\partial\theta}\right|_{\theta=\nu}. \ Further, if$ $p(\tau|\cdot)$ is a log-concave function, then $\hat{J}_\nu$ is concave for any $\nu$.*

*Proof.* Since each $R(\tau_i)$ is nonnegative, so is the ratio $\frac{R(\tau_i)}{p(\tau_i|\theta_0)}$. The sum of lower bounds being a lower bound, this concludes the proof. □

It is now worth going into more detail on the three parameters of Eq. 4.3:

- $\theta$ is the current value of the parameter we are trying to optimize over;
- $\theta_0$ is the parameter value used to gather samples;
- $\nu$ is the parameter used to create the lower bound. Any value of $\nu$ is valid.

There are two special cases of this bound. First, when $\nu = \theta = \theta_0$, this bound becomes an equality and we recover the policy gradient of Williams (1992). However, this equality only holds for the first update of $\theta$. Second, a more interesting case is $\nu = \theta_0 \neq \theta$. In this case, Eq. 4.3 simplifies and we get

$$\hat{J}_{\theta_0}(\theta) = \frac{1}{N} \sum_i R(\tau_i) \left( 1 + \log \frac{p(\tau_i|\theta)}{p(\tau_i|\theta_0)} \right) \ . \tag{4.4}$$

This bound is used by multiple authors (Dayan & Hinton, 1997; Peters & Schaal, 2007; 2008; Kober, 2014; Schulman et al., 2015a) and has the attractive property that it is tight at the beginning of the optimization since we have $\theta = \theta_0$. When we optimize this bound without ever changing the value of $\nu$, we end up with exactly the PoWER algorithm. However, as we optimize $\theta$, this bound becomes looser and it might be useful to change the value of $\nu$.

This suggest an iterative scheme where, after the optimization of Eq. 4.4 has ended in $\theta = \theta_1$, we recompute the bound of Eq. 4.3 with $\nu = \theta_1$. This yields an iterative version of the PoWER algorithm as described in Algorithm 1.

**The data:** Rewards $R(\tau_i)$, probabilities $p(\tau_i|\theta_0)$, initial parameters $\theta_0$
**The result:** Final parameters $\theta_T$
**for** $t = 1$ **to** $T$ **do**

$$\theta_t = \arg\max_\theta \hat{J}_{\theta_{t-1}}(\theta)$$
$$= \arg\max_\theta \sum_i R(\tau_i) \frac{p(\tau_i|\theta_{t-1})}{p(\tau_i|\theta_0)} \left( 1 + \log \frac{p(\tau_i|\theta)}{p(\tau_i|\theta_{t-1})} \right)$$

**end**

**Algorithm 1:** Iterative PoWER

We recall that PoWER is equivalent to Algorithm 1 but with $T = 1$. As we shall see in the experiments, larger values of $T$ lead to vast improvements.

One can also see that Algorithm 1 performs the same optimizations as the PoWER algorithm where new samples would be gathered at each iteration, with the difference that importance sampling is used rather than true samples. Thus, in the spirit of off-policy learning, we have replaced extra sampling with importance sampling. When, and only when, variance starts to be too high, can we sample from the new distribution.

## 5 CONVEX UPPER BOUND

Lemma 3 requires positive returns. This has two undesirable consequences. First, this can lead to very slow convergence. One can see this by creating a setting where all rollouts lead to a positive return except for one which leads to a return of $-\beta < 0$. To maintain the positivity of the returns, we need to shift all the returns by $\beta$ which does not change the optimal policy using the following transformation: $J(\theta) = \int_{\mathcal{T}} p(\tau|\theta)R(\tau) \ d\tau = \int_{\mathcal{T}} p(\tau|\theta)(R(\tau) + \beta) \ d\tau - \beta$. We may now apply lemma 3 and optimize the following lower bound:

$$J_\nu^\beta(\theta) = \int_{\mathcal{T}} p(\tau|\nu) \left( 1 + \log \frac{p(\tau|\theta)}{p(\tau|\nu)} \right) (R(\tau) + \beta) \ d\tau - \beta \ .$$

Without the rollout with a negative return, the returns would not need to be shifted by $\beta$ and we could have optimized $J_\nu^0(\theta)$ instead. The difference between the two is equal to $J_\nu^\beta(\theta) - J_\nu^0(\theta) = -\beta KL(p(\tau|\nu)||p(\tau|\theta))$ where KL is the Kullback-Leibler divergence, which encourages $p(\tau|\theta)$ to be close to $p(\tau|\nu)$. Hence, one rollout with a negative return would slow down our optimization with a regularization term proportional to that return. A simple heuristic would be to discard such rollouts but we would lose all guarantees about the improvement of the expected return.

Further, the positivity of the returns precludes the use of control variates which are especially useful when the shifted rewards are approximately centered on 0. Thus, it prevents us from benefiting of all the existing techniques based around these control variates which would help reducing the variance.

The positivity assumption is required since we multiply our lower bound with the returns. If, instead, we have convex upper bounds of $p(\tau|\theta)$, then this would provide us with a concave lower bound whenever it is associated with a negative return. Lemma 4 provides such a bound.

**Lemma 4.** *Let*

$$u_\nu(\tau|\theta) = p(\tau|\nu) \exp\left[ (\theta - \nu)^\top \frac{\partial \log p(\tau|\theta)}{\partial \theta}\bigg|_{\theta=\nu} \right] , \tag{5.1}$$

*where $p(\tau|\theta)$ is a log-concave function of $\theta$. Then $u_v(\tau|\theta)$ is a convex function of $\theta$ and we have:*

$$u_\nu(\tau|\nu) = p(\tau|\nu) \quad , \quad \frac{\partial u_\nu(\tau|\theta)}{\partial \theta}\bigg|_{\theta=\nu} = \frac{\partial p(\tau|\theta)}{\partial \theta}\bigg|_{\theta=\nu} \quad , \quad u_\nu(\tau|\theta) \geq p(\tau|\theta) \ \forall \theta .$$

*Proof.* Both equalities can be verified by setting $\theta = \nu$. Since $u_\nu$ is the exponential of a linear function in its argument, it is convex. Finally, using the concavity of $\log p$, we have:

$$\log p(\tau|\theta) \leq \log p(\tau|\nu) + (\theta - \nu)^\top \frac{\partial \log p(\tau|\theta)}{\partial \theta}\bigg|_{\theta=\nu}$$

$$p(\tau|\theta) \leq p(\tau|\nu) \exp^{(\theta-\nu)^\top \frac{\partial \log p(\tau|\theta)}{\partial \theta}\big|_{\theta=\nu}} .$$

This concludes the proof. □

We may now combine the two bounds to get the following lower bound on $\hat{J}(\theta)$ without any constraint on the rewards:

$$\hat{J}(\theta) \geq \frac{1}{N} \sum_i R(\tau_i) \frac{p(\tau_i|\nu)}{p(\tau_i|\theta_0)} z_i(\theta) \tag{5.2}$$

$$z_i(\theta) = 1_{R(\tau_i)\geq 0} \left( 1 + \log \frac{p(\tau_i|\theta)}{p(\tau_i|\nu)} \right) + 1_{R(\tau_i)<0} \exp\left[ (\theta - \nu)^\top \frac{\partial \log p(\tau|\theta)}{\partial \theta}\bigg|_{\theta=\nu} \right] .$$

Further, when $p(\tau|\theta)$ is log-concave in $\theta$, then $z_i(\theta)$ is concave in $\theta$.

This allows us to deal with positive and negative rewards, which means that the choice of control variate is now free, which can help both in reducing the variance and improving the lower bound and thus the convergence.

## 6 EXPERIMENTS

To demonstrate the effectiveness of Iterative PoWER, we provide results obtained on the Cartpole benchmark using the Gym [d] toolkit. We also show experiments on real online advertising data where we try to maximize advertisers' revenue while keeping our costs constant.

### 6.1 CARTPOLE BENCHMARK

To show how we can achieve good performance with limited rollouts, we ran our experiments on the Cartpole benchmark, which is solved by the PoWER method. We used a stochastic linear policy with a 4-dimensional state (positions and velocities of the cart and the pole) where, at each time step, the cart moves right with probability $\sigma(s^\top \theta)$ where $s$ is the vector representation of the state. Each experiment was 250 rollouts, in 10 batches of 25 rollouts, of length 400. In between each batch, we retrained the parameters of the model. The average performance over 350 repetitions was computed every batch, varying the number $T$ of iterations of PoWER [e]. We capped the importance weights at 20.

---

[d] https://gym.openai.com/
[e] Each iteration of PoWER consisted in 5 steps of Newton method

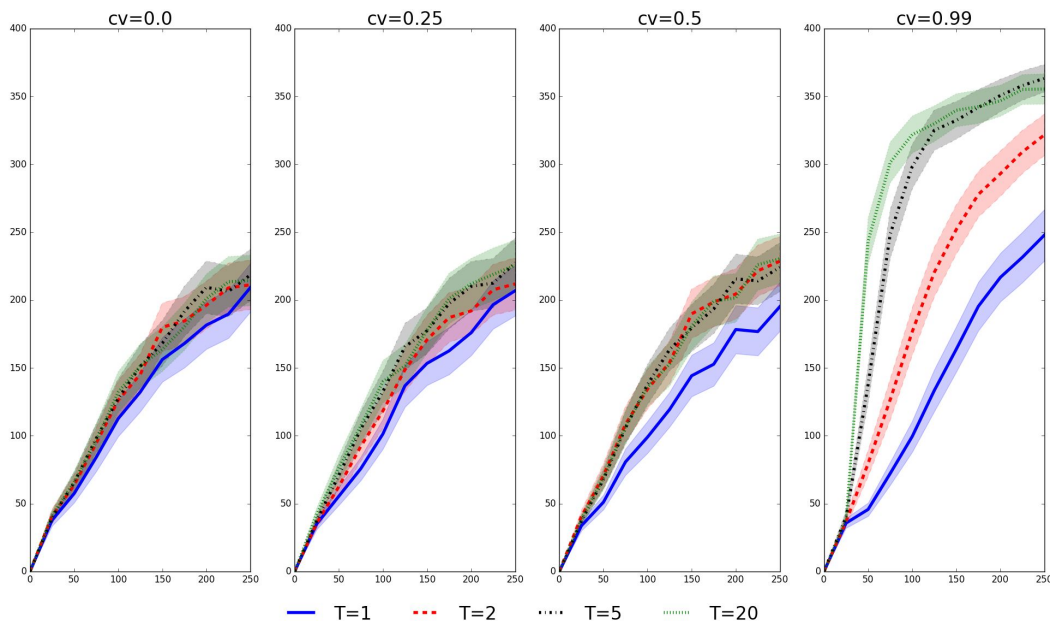

Figure 1: Reward as a function of the number of rollouts for the Cartpole problem, for various values of iterations of Power and various values of the control variate. The light areas represent 3 standard deviations. The performance greatly increases with higher values of $T$, especially with a proper control variate. Except at the very beginning, there is no major difference between $T = 5$ and $T = 20$, leading us to think that we might suffer from variance issues despite the control variates.

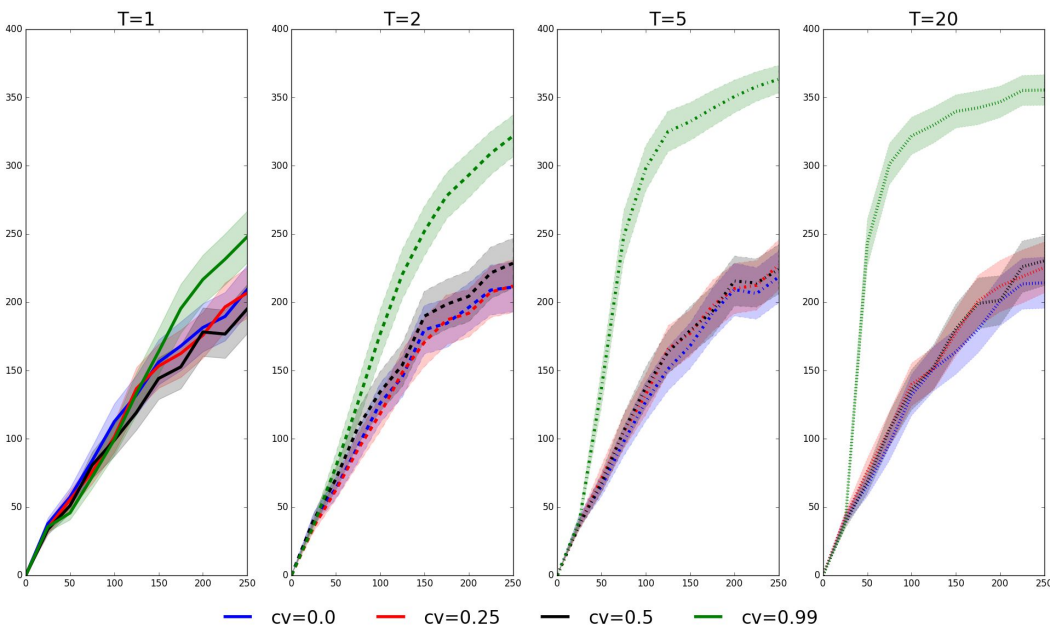

Figure 2: Reward as a function of the number of rollouts for the Cartpole problem, for various values of iterations of Power and various values of the control variate. The light areas represent 3 standard deviations. Using a control variate drastically improves the performance when using large values of $T$ while it has very little impact for $T = 1$ and $T = 2$. This demonstrates the importance of being able to deal with negative rewards.

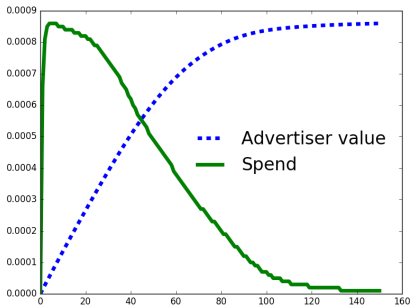

Figure 3: **Dashed blue:** improvement in expected merchant value as a function of the number of iterations (arbitrary linear scale). The optimization is very slow at the beginning and the improvement is close to linear for the first 50 iterations.
**Solid green:** Relative change of the expected cost as a function of the number of iterations. The expected spend increased up to $0.0086\%$ but reached $0.0001\%$ when we stopped the optimization.

We also evaluated the impact of using control variates on the convergence. To that effect, at each iteration, we computed the control variate minimizing the variance of the total estimator. We then used as control variate fractions of that value. For instance, $cv = 0.5$ means that the control variate used was half of the "optimal" control variate found by minimizing the variance of the estimator. The results can be seen in Fig. 2 and Fig. 1. We see that doing several iterations of PoWER leads to an increase in performance, up to a point where the variance is too high.

We can also see that, without using control variates ($cv = 0$), it is not beneficial to do too several iterations. The stagnation in performance for $cv = 0.25$ and $cv = 0.5$ might be due to the poorer quality of the upper bound compared to the lower bound. Thus, to get the full benefit of the control variate, we need to use larger values, such as $cv = 0.99$, which yields the highest performance for any value of $T$.

## 6.2 REAL WORLD DATA - ONLINE ADVERTISING

We also tested Iterative PoWER using real data gathered from a production system. In this system, every time a user lands on a webpage, an auction is run to determine who gets to display an ad to that user. Each participant to the auction chooses in real-time how much to bid for the right to display that ad. If the ad is clicked and then the user buys an item on the merchant's website, a reward equal to the value of the item is observed. Otherwise, a reward of 0 is observed. The cost of displaying an ad depends on the bid in a way unknown to us as we do not have access to the type of auction nor to the bids of the other participants.

We gathered data for over 1.3 billion displays over a period of 2 weeks in April 2016. This data was comprised of information available at the time of the bid, for instance the history of the user, the current URL, or the size of the ad to display. The aggregation of all this data represented the states. When we won the auction, we also logged our bid, which are our actions, the cost of the display and the value generated if there was a sale, both of which are used for the reward.

Rather than learning a full bidding strategy mapping from states to a bid, we used our production system as baseline and learnt a small modifier to account for the non-truthfulness of the auctions. Thus, only a few thousand parameters were learnt, a small number compared to the number of training samples. This allowed us to run many iterations of PoWER without fear of high variance.

Since our aim was to maximize the value generated, this could lead to the undesirable solution where all ads are bought regardless of their price. Thus, we included the constraint that the total cost of buying the ads had to remain constant. The details of iPoWER with added constraints are in section 7.1

Fig. 3 shows the relative improvement in advertiser value generated as a function of the number $T$ of iterations. We can readily see that the final improvement obtained by Iterative Power is far greater than that obtained after one iteration of PoWER (about 60 times greater). The green curve shows the change in total cost. Since we also used a lower bound for the constraint, it is initially not satisfied, but running the algorithm to convergence leads to a solution in the feasible set. The final improvement represents an increase of the net gain by several percentage points.

## 7 EXTENSIONS AND CONCLUSION

### 7.1 APPLICATION TO CONSTRAINED OPTIMIZATION

There are cases where one wishes to maximize the expected reward while satisfying a constraint on another quantity. In online advertising, for instance, it is interesting to maximize the number of clicks (or sales) while keeping the number of ads displayed constant as this reduces the potential long-term effects on user behaviour, something not captured at the scale of a rollout. To maximize the expected reward $\mathbb{E}_\theta[R]$ while satisfying the constraint $\mathbb{E}_\theta[S] = S_0$, we may add a Lagrangian term to $\hat{J}_\nu(\theta)$ and iteratively solve the following problem:

$$\max_\theta \min_\alpha \sum_i R(\tau_i) \frac{p(\tau_i|\nu)}{p(\tau_i|\theta_0)} \left(1 + \log \frac{p(\tau_i|\theta)}{p(\tau_i|\nu)}\right) + \alpha \left(\sum_i S(\tau_i) \frac{p(\tau_i|\nu)}{p(\tau_i|\theta_0)} \left(1 + \log \frac{p(\tau_i|\theta)}{p(\tau_i|\nu)}\right) - S_0\right)$$

where $\alpha$ is the Lagrange multiplier associated with the constraint. Due to the approximation, the constraint will not be exactly satisfied for the first few iterations but the convergence of this algorithm guarantees that the constraint will eventually be satisfied.

### 7.2 EXTENSION TO NON LOG-CONCAVE POLICIES

The results of this paper rely on a log-concavity assumption on the policy which can be too strong a constraint. Indeed, in many cases, the policy depends in a complex manner on the state, for instance through a deep network. However, most of these policies can still be written as a log-concave policy on a non-linear transformation of the states, for instance when the last layer of the deep network uses a softmax. iPoWER can then be used to transform the optimization problem into a simpler, albeit still not concave, maximization problem where the non-concavity of the output of the network has been removed and only remains the non-concavity of the nonlinear transformation of the state.

### 7.3 CONCLUSION

We proposed a modification to the PoWER algorithm which allows to improve a policy with a reduced number of rollouts. This method is particularly interesting when there are constraints on the number of rollouts, for instance in a robotic environment or when each policy has to be deployed in an industrial production system. We also proposed an extension to existing EM-based methods which allows for the use of control variates, a potentially useful tool to reduce the variance of the estimator. However, several questions remain. In particular, experiments on the Cartpole benchmark indicate that, despite the use of capped importance weights and control variates, as we do more iterations, we might end up in regions of the space with high variance. It is thus important to use additional regularizers, such as normalized weights or penalizing the standard deviation of the estimator. To maintain the simplicity of the overall algorithm, concave lower bounds of these regularizers must also be found, which is still an open problem.

ACKNOWLEDGMENTS

The author would like to thank Vianney Perchet for helpful discussions.

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
