# Peer review of "Efficient iterative policy optimization"

_ICLR 2017 — rejected_

[Official Review · AnonReviewer3 · rating 7 · confidence 3 · 16 Dec 2016]

The paper considers the problem of reinforcement learning where the number of policy updates is required to be low. The problem is well motivated and the author provides an interesting modification to the PoWER algorithm, along with variational bounds on the value function (lemmas 3,4) which are interesting in themselves. They also provide numerical results on the cartpole problem and a problem in online advertising with real data. Overall this is a strong, well-written paper. My main reservation is whether it is completely appropriate for ICLR, since the log-concavity assumption the model relies on appear to restrict to simpler models where representations will be not in fact be learned.

Other comments:
- There is a general lack of baselines in the numerical experiment section. I acknowledge this is somewhat of an unusual setting, but even a simple, well-justified baseline would have been welcome. Since cartpole is a relatively simple problem and the advertising dataset is presumably private, perhaps a way to generate a synthetic advertising dataset would have been interesting.

- I was confused by the control variates as constant scalars - are they meant to be constant baselines? And if so, they appear to be treated as hyperparameters -  why are they not learned or estimated?

- There is an interesting section on constrained optimization, but as it is, feels a bit disconnected from the rest of the paper. It appears applicable to the problem of online advertising, but is not mentioned in the corresponding experimental section. Also might be worth adding a citation to the literature of constrained MDPs which develops similar lagrangian ideas.

[Official Review · AnonReviewer1 · rating 5 · confidence 4 · 16 Dec 2016]

The paper presents an interesting modification to PoWER algorithm that is well motivated. The main limitation of this paper is the lack of comparison with other methods and on richer problems. The experiments haven't given confidence to show its claimed benefits, generality and scalability over prior methods. Giving this confidence doesn’t necessarily require running your method on all large-scale domains or doing exhaustic hyper-parameter search, but for example it could go beyond current domains. Cartpole only optimizes 5 parameters. Ad targeting task lacks comparison with alternative methods. Since this method is built on PoWER and closely connected with RWR, it is likely there are limits to performance which may become apparent when the method is tried on other domains and with other benchmark methods, e.g. even standard ones like importance sampling-based off-policy learning is known to suffer in high-dimensional or continuous action space; limits of RWR/PoWER-like methods based on their connection with entropy-regularized RL.

[Official Review · AnonReviewer2 · rating 3 · confidence 2 · 20 Dec 2016]
**iterative PoWER and control variates**

This paper presents iterative PoWER, an off-policy variation on PoWER, a policy gradient algorithm in the reward-weighted family.

I'm not familiar enough with this type lower bound scheme to comment on it. It looks like the end result is less conservative step sizes in policy parameter space. All expectation-based algorithms (and their KL-regularized cousins a-la TRPO) take smallish steps, and this might be a sensible way to accelerate them.

The description of the experiments in Section VI is insufficient for reproducibility. Is "The cart moved right" supposed to be "a positive force is applied to the cart"? How is negative force applied? What is the representation of the state? What is the distribution of initial states? A linear policy is insufficient for swing up and balance of a cart-pole. Are you only doing balancing? What is the noise magnitude of the policy? How was it chosen? How long were the episodes?

The footnote at the bottom of page 8 threw me off. If you're using Newton's method, where is the discussion of gradients and Hessians? I thought the argmax_theta operator was a stand-in for an EM-style step, which I how I read Eq (8) in the Kober paper.

[Final Decision · Program Chairs · 06 Feb 2017]
**ICLR committee final decision**

The reviewers generally agreed that exploring policy search methods of this type is interesting, but the results presented in the paper are not at the standard required for publication. There are no comparisons of any sort, and the only task that is tested is trivially simple, so it's impossible to conclude anything about the effectiveness of the method. Despite the author promising to add additional experiments, nothing was added in the current draft. Besides this, reviewers raised concerns about the relevance of this approach to ICLR. The crucial point here is that it's unclear if the method will scale -- while there is nothing wrong in principle in proposing a general policy search method, there isn't really a compelling argument that can be made that it is suitable for learning representations if there is no plausible story for how it will scale to sufficiently complex policy classes that can actually learn representations.